# Numerical Simulation Study on Environment-Friendly Floating Reef in Offshore Ecological Belt under Wave Action

**Yun Pan** [1] , **Huanhuan Tong** [2], **Yang Zhou** [2,*], **Can Liu** [2] and **Dawen Xue** [1,*]

1   School of Naval Architechture and Marinetime, Zhejiang Ocean University, Zhoushan 316022, China; panyunhk@zjou.edu.cn
2   School of Marine Engineering Equipment, Zhejiang Ocean University, Zhoushan 316022, China; tongzjou@163.com (H.T.); liucanzjut@126.com (C.L.)
*   Correspondence: edit502@126.com (Y.Z.); xuedw@zjou.edu.cn (D.X.)

**Abstract:** An artificial floating reef is an important part of the coastal ecological corridor. The large-scale construction of floating reefs by optimizing mooring methods can effectively improve the ecological effects of coastal projects. The artificial floating reef belongs to coastal engineering, and wave resistance is fundamental to its structural design. In this paper, the method for processing coupling forces and motion, the method for judging the floating reef out of water surface, and the method for correcting velocity and acceleration of water mass points are elaborated in detail by using the finite element method and lumped-mass mooring model. By comparing and analyzing the results of physical experiment and numerical simulation, the correctness of the numerical model is verified. Finally, the diachronic variation of pitching angle of floating reef, the tension of the mooring rope, and the total tension of the fixed points of the fishing net were analyzed by the dynamic response numerical mode with a new type of mooring. The purpose of the current study was to provide a basis for the optimization of structure shape, the matching of floating body, and the counterweight of artificial floating reef.

**Keywords:** coastal ecological corridor; artificial floating reef; wave action; dynamic response; mooring type

## 1. Introduction

A coastal ecological corridor [1,2] is a new thing that China has proposed in recent years, including coastal green belts adjacent to land areas, seawall ecological belts, and coastal beach ecological belts. Its functional positioning is mainly to protect biodiversity, filter pollutants, purify water bodies, weaken waves, protect against erosion, and enhance landscape effects, thereby improving the coastal ecological environment. Coastal ecological corridor projects include biological protection projects and ecological coastal engineering structures. In addition, the structural design of ecological coast engineering includes typical structural types and additional ecological structures. Floating reef is an important part of the additional ecological structure [3]. They can further carbonize, adsorb, and degrade pollutants by attaching algae and shellfish to optimize water quality and restore ecology. The *Technical Guidelines for Ecological Construction of Reclamation Projects (Trial)* issued by China State Oceanic Administration in 2017. It was pointed out that the use of artificial reefs and other ecological designs can provide breeding, growth, bait, and shelter for fish, shellfish, etc., and create a good environment for marine life to inhabit. In addition, according to the *National Marine Ranch Demonstration Zone Construction Plan (2017–2025)* issued by China Ministry of Agriculture, by 2025, it is planned to build an artificial reef area of 160 square kilometers in the East China Sea district, and put more than 5 million empty cubic meters of artificial reef. To sum up, it can be seen that China attaches great importance to the construction of a floating reef in coastal areas. However, the selection of the type of floating reef, the distribution of the floating reef's place, the safety of the structure and

materials, and the effect of fish aggregation and shelter are important prerequisites for the scientific creation of fish spawning and conservation grounds.

Japan is the first country to study and use the artificial floating reef. It is pointed out that artificial floating reef can be defined as a systematic process which consists of three elements or components: (1) buoyancy system, (2) reef system, (3) mooring system. Firstly, buoyancy system components are composed of a float tube and float ball. It cannot only provide buoyancy for the floating reef, but also serve as the main source of wave loads. Secondly, the reef system is comprised of the rigid steel or the plastic frame covered with the fishing net, the plastic membrane, and the canvas. The fishing net has a better permeability and flexibility and it is easy to attach algae organisms. Therefore, it is sure that the fishing net can create a spawning place and bait area for fish. Thirdly, the mooring system has a great influence on the motion characteristics of floating reef, so the moorage of the floating body, the coordination between each system, and the collocation between components need optimal buoyancy and counterweight. Last but not least, it is very important for the design and deployment of artificial floating reef to possess safe structure, stable performance, coordinated system, and anti-wave ability.

In recent years, Japan [4,5] and South Korea [6] have designed and put typical structure type of artificial floating reef into offshore. Furthermore, a new structural type of floating reef, which is used for eco-coastal projects, are put into sea with a few monomers and evaluated with fish aggregation. However, there are relatively few historical studies in the area of marine hydrodynamic properties, which include the safety, stability, deployment of floating reefs, and other fundamental engineering problems or techniques. The reason why the main body of the reef is very similar to the structure of aquaculture cage includes two aspects: (1) the buoyancy system of floating reef includes float tube and float ball, which based on permutation and combination; (2) the reef's body is woven by mooring rope and fishing net. (Hou et al. [7]; Huang et al. [8]; Bai et al. [9]; Zhao et al. [10]; Gui et al. [11]; Moe-Føre et al. [12]; Strand et al. [13]) have long been committed to the research of the structure type of net cage, the movement of net cage's floating frame, the mode of mooring, and the change of net cage's volume. They solved the problem of the size reduction and structural safety of the flexible structure of net cage under the action of waves and tidal currents, and ensured that net cage could meet the design requirements in different marine environments. All of these provide a safeguard for the study of the dynamic response of floating reefs under wave action. The artificial floating reef, which is different from the net cage's structure, also needs to maintain a certain height and attitude in order to act on the shoal of fish in the surrounding waters [4]. With vertical dimension 5–10 times larger than horizontal dimension, artificial reefs can be deployed in groups and it is usually tied by single mooring [4–6]. The difference of numerical modeling approach between net cage and artificial floating reef are the calculation of the rotational inertia, the Blaine angle of rotational transformation of complex floating structures, and the treatment of the mutual coupling of forces and motions at the points or units where the constituent components are connected, which is the focus of this research.

## 2. Establishment of Numerical Model

The components of the floating reef are connected by thermal fusion and rope tying, so the key to the numerical modeling of the overall reef movement is the treatment of the points or units at the location of the connections. This section introduces the numerical calculation of the force and motion of the components such as the circular tube floating frame, fishing net, and mooring rope under the action of wave, and then focuses on the treatment methods of the coupling force and motion among the three. Numerical simulation methods for the motion of the floating frame, fishing net, and mooring rope under wave action are divided into two cases: rigidity and flexibility. The floating frame is a rigid structure, and the motion includes translation and rotation, while the fishing net and mooring rope are flexible structures, and the form of motion are tension and suspension. The numerical model is divided into three steps as follows. Firstly, it mainly

uses Morrison equation to calculate the forces on the dividing unit or the concentrated mass points. Secondly, the resultant force of the whole structure and the torque around the center of mass is achieved. Finally, the force change and motion state of the structure are obtained through discrete time.

### 2.1. Force and Movement of Circular Tube Floating Frame

The floating frame adopts HDPE piping fusion welding. In a small scale, the pipe can be considered as rigid and can maintain the design shape. As the velocity and acceleration of water mass point along the direction of water depths gradually decrease, horizontal wave force on the upper part of floating body centered on the center of mass is greater than that on the bottom, which will lead to the rotation of the floating reef body. This phenomenon can be accurately described by calculating the wave forces of the finite elements of floating frame after they are divided by finite element.

#### 2.1.1. Force and Unit Division of Floating Frame

The main forces of circular tube floating frame subjected to waves include gravity, buoyancy, wave force, tension of fishing net, and mooring rope. When the ratio of diameter of circular tube to the wavelength is less than 0.2, wave force can be calculated using Morrison equation. The calculation of wave force which include drag force and inertial force and the selection of hydrodynamic coefficient of moving circular tube are described clearly in reference [14–16]. When the floating frame rotates, the angle between the velocity direction of water mass point or the acceleration direction of water mass point and the circular tube changes, and a local coordinate system needs to be established on the circular tube. For details, please refer to literature [14–16]. Different from those literature, the 3D floating frame movement model is established and fishing net is wound around it, as shown in Figures 1 and 2. Taking the front face ABCD of the floating frame as an example, in order to ensure that the mesh of fishing net is completely tied to the tube, the circular tube's unit division of the horizontal direction AB and CD or the vertical direction AD and BC needs to meet the size of the mesh. Guo et al. [17], Wan et al. [18] usually adopted the mesh shape with the horizontal hanging ratio of 0.66 and the vertical hanging ratio of 0.75, which are the ratio of the length of the horizontal and vertical diagonal to the length of the mesh bar. If the mesh grouping technique [17,18] adopts 8 × 8, the horizontal length of mesh is 0.4224 m and the vertical length is 0.48 m. Figure 1 shows the way of units dividing of the front face of floating frame. As the mesh is attached to the odd units of the circular tube, the length of the horizontal unit is 0.2112 m and the length of the vertical unit is 0.24 m.

In Figure 1, each square represents the lumped point of the physical quantity of half the unit length. These physical quantities include the mass of unit, water mass point velocity, water mass point acceleration, and the fulcrum to calculate the force arm, etc. Therefore, except for three adjacent unit nodes at the four endpoints in the 3D model, such as A, B, C, and D, the remaining unit nodes are only two adjacent unit nodes at the left and right sides.

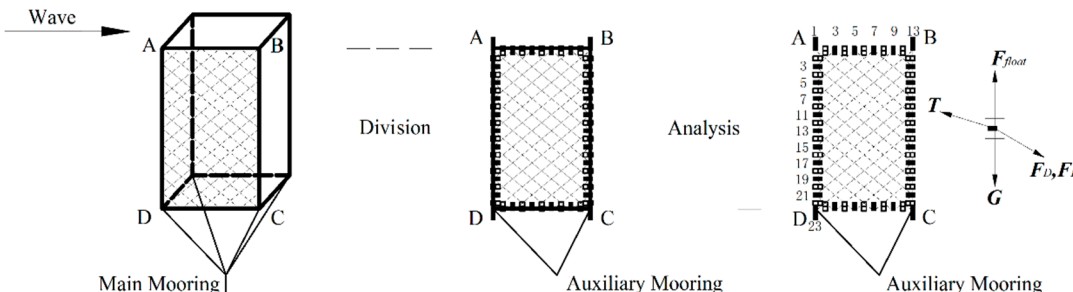

**Figure 1.** Unit division and force analysis of the floating frame.

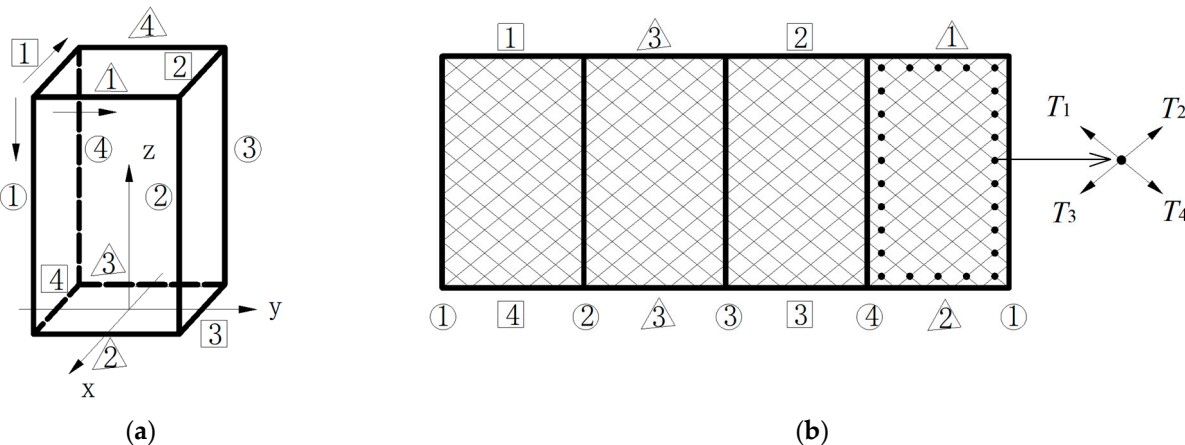

**(a)**　　　　　　　　　　　　　　　　　　　　　　**(b)**

**Figure 2.** Calculation method of the floating structure and fishing net. (**a**) Floating frame's structure number (Circular label, Square label, Triangle label). (**b**) Numbering method of fishing net and force on the node of fishing net.

In model, the origin of overall coordinate is established on the central axis of floating frame. $z = 0$ coincides with the stationary water surface and the positive direction of $x$, $y$, and $z$ axes is shown in Figure 2a. The initial position coordinates of a given floating frame are divided into three cases: Firstly, No. 1~4 circular label circular tubes parallel to the $z$ axis in Figure 2a. Array of Numbers are written by calculating program from top to bottom direction, such as $P1$ ($i_{bar}$, 4, 3). $i_{bar}$ represents divide such circular tubes into $i_{bar}$ units; No. 4 represents the number of this type of circular tube and No. 3 represents the values of 3D global coordinates in $x$, $y$ and $z$ direction. Secondly, No. 1~4 Square label circular tubes parallel to the $x$ axis in Figure 2a. The direction of array number is from front to back, such as $P2$ ($j_{bar}$, 4, 3). $j_{bar}$ indicates divide such circular tubes into $j_{bar}$ units. Thirdly, No. 1~4 Triangle label circular tubes parallel to the $y$ axis in Figure 2a. The direction of array number is from left to right, such as $P3$ ($j_{bar}$, 4, 3), which have the same numerical meaning as above.

In this model, how to establish a local coordinate system ($\xi$, $\eta$, $\tau$) on each circular tube that composes the floating frame is very important. The direction of $\tau$ is along the direction of the circular tube. The $\xi$ axis, which is on the plane formed by $\tau$ and the relative velocity $\mathbf{V}_r$ of water particles, is perpendicular to $\tau$. In the global coordinate system, taking $P1$ ($i_{bar}$, 4, 3) as an example, the unit vector of the local coordinate system ($\xi$, $\eta$, $\tau$) can be obtained by the relative velocity $\mathbf{V}_r = \mathbf{V}_{water} - V1(i_{bar}, 4, 3)$ and the $\tau = P1$ ($i_{bar} + 1$, 4, 3) $- P1$ ($i_{bar}$, 4, 3) with the method of cross product. The $\mathbf{V}_{water}$ is the velocity of the water quality point at the position of $P1$ ($i_{bar}$, 4, 3) in the regular wave field as is shown in Equation (1), and $V1(i_{bar}, 4, 3)$ is the velocity vector of the dividing point of $P1(i_{bar}, 4, 3)$. The calculation method of wave force is shown in Equation (2) to (4).

$$\begin{cases} V_{water-x} = \frac{\pi H_{wave}}{T_{wave}} \frac{\cosh(k(z+d))}{\sinh(kd)} \cos(kx - \omega t) \\ V_{water-y} = 0.0 \\ V_{water-z} = \frac{\pi H_{wave}}{T_{wave}} \frac{\sinh(k(z+d))}{\sinh(kd)} \sin(kx - \omega t) \end{cases} \tag{1}$$

$$\begin{cases} \mathbf{e}_\xi = \frac{\tau \times \mathbf{V}_r \times \tau}{|\tau \times \mathbf{V}_r \times \tau|} = (x_\xi, y_\xi, z_\xi) \\ \mathbf{e}_\eta = \frac{\tau \times \mathbf{V}_r}{|\tau \times \mathbf{V}_r|} = (x_\eta, y_\eta, z_\eta) \\ \mathbf{e}_\tau = \frac{\tau}{|\tau|} = (x_\tau, y_\tau, z_\tau) \end{cases} \tag{2}$$

$$\begin{cases} \mathbf{F}_{D\xi} = \frac{1}{2}\rho_{water}C_{D\xi}A_\xi \frac{\mathbf{V}_r \cdot \mathbf{e}_\xi |\mathbf{V}_r \cdot \mathbf{e}_\xi|}{2} \\ \mathbf{F}_{D\eta} = \frac{1}{2}\rho_{water}C_{D\eta}A_\eta \frac{\mathbf{V}_r \cdot \mathbf{e}_\eta |\mathbf{V}_r \cdot \mathbf{e}_\eta|}{2} \\ \mathbf{F}_{D\tau} = \frac{1}{2}\rho_{water}C_{D\tau}A_\tau \frac{\mathbf{V}_r \cdot \mathbf{e}_\tau |\mathbf{V}_r \cdot \mathbf{e}_\tau|}{2} \end{cases} , \begin{cases} \mathbf{F}_{I\xi} = \rho_{water}\Lambda C_{M\xi} \frac{\partial \mathbf{V1}}{\partial t} \cdot \mathbf{e}_\xi \\ \mathbf{F}_{I\eta} = \rho_{water}\Lambda C_{M\eta} \frac{\partial \mathbf{V1}}{\partial t} \cdot \mathbf{e}_\eta \\ \mathbf{F}_{I\tau} = \rho_{water}\Lambda C_{M\tau} \frac{\partial \mathbf{V1}}{\partial t} \cdot \mathbf{e}_\tau \end{cases} \tag{3}$$

$$\begin{cases} \mathbf{F}_D = \mathbf{F}_{D\xi} + \mathbf{F}_{D\eta} + \mathbf{F}_{D\tau} \\ \mathbf{F}_I = \mathbf{F}_{I\xi} + \mathbf{F}_{I\eta} + \mathbf{F}_{I\tau} \end{cases} \tag{4}$$

where $\mathbf{e}_\xi$, $\mathbf{e}_\eta$ and $\mathbf{e}_\tau$ are the unit vectors of the local coordinate system, respectively; $\mathbf{F}_D$ and $\mathbf{F}_I$ are combined drag force and inertia force of floating frame, respectively. $C_D$ and $C_M$ are the coefficient of drag force and inertia force, respectively. $\rho_{water}$ represents seawater density. $A_*$ and $\Lambda$ is the force area and volume of the circular pipe of wave front, respectively. In particular, the calculation of the wave force of fishing net and mooring rope, which will be introduced next, also adopts the method of local coordinate system. No more detailed explanation will be given.

2.1.2. Movement of Floating Frame

Fishing net is tied around the floating frame of floating reef. Under the traction of four auxiliary moorings and one main mooring, the movement of the floating frame includes translation and rotation. According to the synthesis principle of velocity and acceleration of rigid body motion, the kinematic velocity and acceleration of rigid body in absolute coordinate system are actually the transport velocity and acceleration, which include translation and rotation.

(1) Calculate Translation:

Regardless of the rotation of floating frame, the forces of all units of floating frame are accumulated in the global coordinate system, as shown in Figure 1. The forces mainly include wave force, net pulling force, buoyancy, and gravity. Then the acceleration of the floating frame is obtained according to Newton's second law, as shown in the following Equation (5):

$$\begin{cases} \mathbf{a}_{frame1} = (\mathbf{T} + \mathbf{F}_D + \mathbf{F}_I + \mathbf{G} + \mathbf{F}_{float})/(M + \Delta M) \\ \mathbf{V}_{frame} = \mathbf{V}_0 + \mathbf{a}_{frame1} \cdot \Delta t \end{cases} \tag{5}$$

where the letter $M$ and $\Delta M$ are total mass of floating frame and the sum of inertial mass of each unit of floating frame, respectively. If the volume of a unit is $\mathbf{V}_{Volume}$ and drag force coefficient is $C_M$, inertial mass is $\Delta M = \rho_{water}(C_M - 1) \cdot \mathbf{V}_{Volume}$. $\mathbf{a}_{frame1}$ is the translational acceleration of floating frame; $\mathbf{T}$ indicates the tension of mooring rope and fishing net on frame, $\mathbf{F}_D$ and $\mathbf{F}_I$ are combined drag force and inertia force of floating frame, respectively; $\mathbf{G}$ is the gravity of floating frame, $\mathbf{F}_{float}$ represents the buoyancy of floating frame, $\mathbf{V}_{frame}$ the translational velocity of floating frame, $\mathbf{V}_0$ is the initial velocity of frame floating body or the velocity at previous moment. In particular, all bold variables in Equation (5) represent vectors in $x$, $y$, and $z$ directions, and the following formulas are the same.

(2) Derivation of the Formula for the Moment of Inertia of Floating Frame:

Correct calculation of the moment of inertia of floating frame is the premise of calculating its rotation. As shown in Figure 2a, the origin of dynamic coordinates of the center of mass of rigid body can calculate rotation. According to the parallel axis theorem, the moment of inertia of 12 circular tubes of floating frame is transferred to the center of mass in $x$, $y$, and $z$ directions. As shown in Figure 2a, the circular tubes that make up the floating frame can be divided into three categories, namely circular label 1, square label 1, and triangle label 1. Assuming that the mass per unit length of the tube is m, the circle label 1 is divided into $i_{bar}$ units, and the length of each unit is $L_{vertical}$. Square label 1 and triangle label 1 are divided into $j_{bar}$ units, and the length of each unit is $L_{lever}$. The expressions for calculating moment of inertia of three types' circular tubes, which transferred to the center of mass of the floating frame, are Equations (6) and (7).

The rotational inertia of circular label 1 in $x$, $y$, and $z$ directions are shifted to the center of mass:

$$\begin{cases} I_{x1} = (i_{bar} - 1)L_{vertical}m \cdot [(i_{bar} - 1)L_{vertical}]^2/12 + (i_{bar} - 1)L_{vertical}m \cdot [(j_{bar} - 1)L_{lever}/2]^2 \\ I_{y1} = I_{x1} \\ I_{z1} = (i_{bar} - 1)L_{vertical}m \cdot r_{bar}^2/2 + (i_{bar} - 1)L_{vertical}m \cdot 2[(j_{bar} - 1)L_{lever}/2]^2 \end{cases} \tag{6}$$

The rotational inertia of square label 1 in the $x$, $y$, and $z$ directions are shifted to the center of mass:

$$\begin{cases} I_{x2} = (j_{bar} - 1)L_{lever}m \cdot r_{bar}{}^2/2 + (j_{bar} - 1)L_{lever}m \cdot \left\{ [(i_{bar} - 1)L_{vertical}/2]^2 + [(j_{bar} - 1)L_{lever}/2]^2 \right\} \\ I_{y2} = (j_{bar} - 1)L_{lever}m \cdot [(j_{bar} - 1)L_{lever}]^2/12 + (j_{bar} - 1)L_{lever}m \cdot [(j_{bar} - 1)L_{lever}/2]^2 \\ I_{z2} = I_{y2} \end{cases} \tag{7}$$

The rotational inertia of triangle label 1 in the $x$, $y$, and $z$ directions are shifted to the center of mass, respectively: $I_{x3} = I_{y2}$, $I_{y3} = I_{x2}$, $I_{z3} = I_{y3}$. Therefore, the total rotational inertia of floating frame in $x$, $y$, and $z$ directions are: $I_x = 4(I_{x1} + I_{x2} + I_{x3})$, $I_y = 4(I_{y1} + I_{y2} + I_{y3})$, $I_z = 4(I_{z1} + I_{z2} + I_{z3})$.

(3) Calculate Rotation:

The rotation of rigid body is relative to the origin of global coordinate system and the center of mass of rigid body of global coordinate system is usually taken as the origin of moving coordinate system for rotational calculation. The specific calculation methods and steps of rotation are as follows:

$$\mathbf{M} = \sum_{i=1}^{n} \mathbf{F}_{total}^i \times \mathbf{R} \tag{8}$$

where $\mathbf{M}$ is the total torque of frame type floating body, $\mathbf{F}_{total}$ is join forces on the unit $i$; $\mathbf{R}^i$ is the arm of the unit rotating with respect to the center of mass; $\times$ is the cross product of vectors.

$$\begin{cases} \mathbf{a}_{frame2} = \mathbf{M}/\mathbf{I} \\ \mathbf{w}_{frame} = \mathbf{w}_0 + \mathbf{a}_{frame2} \cdot \Delta t \\ \theta = \mathbf{w}_{frame} \cdot \Delta t + 0.5 \cdot \mathbf{a}_{frame2} \cdot \Delta t^2 \end{cases} \tag{9}$$

where $\mathbf{I}$ is the moment of inertia of floating frame; $\mathbf{a}_{frame2}$ represents angular acceleration around the center of mass of floating frame, $\mathbf{w}_{frame}$ represents angular velocity around the center of mass of floating frame; $\mathbf{w}_0$ represents the initial velocity of scaffold or angular velocity at the last moment, $\theta$ is the rotation angle of float frame in $\Delta t$ time.

According to the calculation formula of Blaine angle [10,11], it can be known that the angle of rotation of the frame floating body within $\Delta t$ in the three-dimensional situation is equal to the Blaine angle at that moment. The specific rotation and dynamic synthesis of the floating frame, namely the transformation formula of the moving coordinate system and the overall coordinate system, is as follows:

$$\begin{cases} \mathbf{V}^i = \mathbf{V}_{frame1} + \mathbf{w}_{frame} \cdot \mathbf{R}^i \\ P_x^i = R_x^i \cdot \cos\theta_y \cos\theta_z + R_x^i \cdot \cos\theta_y \sin\theta_z + R_z^i \cdot \sin\theta_y + P_x^{center} \\ P_y^i = R_x^i \cdot (-\sin\theta_x \sin\theta_y \cos\theta_z - \cos\theta_x \cos\theta_z) + R_y^i \cdot (-\sin\theta_x \sin\theta_y \sin\theta_z + \cos\theta_x \cos\theta_z) \\ \quad + R_z^i \cdot \sin\theta_y \cos\theta_y + P_y^{center} \\ P_z^i = R_x^i \cdot (-\cos\theta_x \sin\theta_y \cos\theta_z + \sin\theta_x \sin\theta_z) + R_y^i \cdot (-\cos\theta_x \sin\theta_y \cos\theta_z - \sin\theta_x \cos\theta_z) \\ \quad + R_z^i \cdot \cos\theta_y \cos\theta_y + P_z^{center} \end{cases} \tag{10}$$

where $\mathbf{V}^i$ is the velocity in the overall coordinate system of the $i$-th unit of the floating frame. Since each unit $\mathbf{V}_{frame}$ and $\mathbf{w}_{frame}$ are equal, there is no need to mark with $i$. $P_*^i$ is the coordinates of $x$, $y$, and $z$ in the global coordinate system of unit $i$ of the floating frame, which are successively $P_x{}^i$, $P_y{}^i$ and $P_z{}^i$, $P_*^{center}$ is the coordinate of the moment in the global coordinate system of the center of mass of floating frame; $\theta_x$, $\theta_y$, $\theta_z$ represent the Brian Angle of the $x$, $y$, and $z$ axes, respectively.

### 2.2. Tension and Movement of Fishing Net

Assuming that the fishing net is composed of lumped mass points connected by a limited massless spring, the force and shape of the net can be obtained by calculating the displacement of the lumped mass point under the action of waves, buoys, and mooring ropes [17,18]. The lumped mass points of fishing net are set at both ends of each mesh bar, and each lumped mass point contains 1 mesh knot and 2 mesh bars. As shown in Figure 2b,

among them, the tension of four netting twine at any lumped point is named $T_1 - T_4$ in clockwise direction. For the tension and movement of single fishing net, please refer to the literature [17,18], which provides detailed introduction of mesh clustering technology, lumped mass point method, and mesh's local coordinate establishment, etc.

The following mainly introduces the processing method of the mesh bar of fishing net how just completely tied around the floating frame and the calculation method of the netting twine tension at the tie points. Figure 2b shows the result of the floating frame unfolded along the circular label 1. For the convenience of modeling and the idea of using a piece of fishing net, the storage variable form of fishing net is $P_{net}(i_{bar}, 1 + 4(j_{bar} - 1, 3)$. Since there are three duplicates at the connecting ends of the square label and the triangle label tube, the number of columns of fishing net should be $1 + 4(j_{bar} - 1)$.

In order to make all mesh nodes, which are referred to as mass lumped points, we use the same number and tension of netting twine. In Figure 2b, the mass is not concentrated at the binding points, but at the solid dots. Therefore, among these points, the endpoints of four corners concentrated 3.5 times single fishing net's twine mass and the other 3 times. Accordingly, the binding points on the circular tube parallel to the $x$ and $y$ axes are pulled by adjacent fishing net's nodes, namely $T_1$ and $T_2$, while the binding points parallel to the $z$ axis are pulled by adjacent fishing net's nodes, namely $T_1$ and $T_3$, $T_2$ and $T_4$. According to the coordinates of the tie points and the coordinates of the mesh nodes, the vector sum of the force on the tie points can be obtained by decomposing netting twine's tension.

As shown in Figure 2b, serial number and the units' quantity of ($j_{bar}$) of circular tubes parallel to the $x$ and $y$ axes are different from those of the fishing net. The transformation relationship between them is given in Equation (11), which is the key to calculate the connection points (binding points) between fishing net and floating frame.

$$\begin{cases} j_{p2} = j, j = 1, 3, \ldots 1 + (j_{bar} - 1) \\ j_{p3} = j - (j_{bar} - 1), j = 1 + (j_{bar} - 1), 3 + (j_{bar} - 1), \ldots 1 + 2(j_{bar} - 1) \\ j_{p2} = j - [(j + 1)/2 - (j_{bar} - 3)] * 4, j = 1 + 2(j_{bar} - 1), 3 + 2(j_{bar} - 1), \ldots 1 + 3(j_{bar} - 1) \\ j_{p3} = j - [(j + 1)/2 - j_{bar}] * 4, j = 1 + 3(j_{bar} - 1), 3 + 3(j_{bar} - 1), \ldots 1 + 4(j_{bar} - 1) \end{cases} \quad (11)$$

where $j_{P2}$ and $j_{P3}$ correspond to the serial number of square label tube and triangle label tube in turn, $j$ is the column number of any fishing nets, and its value is 1~1 + 4 ($j_{bar} - 1$).

### 2.3. Tension and Movement of Mooring Rope

The numerical simulation method of mooring rope is relatively mature. The method of lumped mass point is adopted to classify mooring rope into sphere points with massless spring links, which can better describe the force and movement of the mooring rope. Different from the tension of spring, the mooring rope is only stressed when it is longer than the original length, as shown in Equation (12). The tension and calculation of lumped mass point of mooring rope are similar to fishing net's twine, mainly including gravity, buoyancy, tension of mooring rope, wave drag force and wave inertia force, etc. The specific calculation formula is shown in the equation and Equation (12).

$$\begin{cases} \mathbf{a}_{mooring} = (\mathbf{T}_{mooring} + \mathbf{F}_{D-mooring} + \mathbf{F}_{I-mooring} + \mathbf{G}_{mooring} + \mathbf{F}_{float-mooring}) / (M_{mooring} + \Delta M_{mooring}) \\ T_{mooring} = d^2 + C_1 \varepsilon^{C_2}, \varepsilon = (l - l_0)/l_0 \end{cases} \quad (12)$$

where $M_{mooring}$ is the mass of mooring rope's mass point, $\Delta M_{mooring}$ indicates inertial mass increment of mooring rope's mass point, $\mathbf{a}_{mooring}$ the motion acceleration of mooring rope's mass point; $\mathbf{T}_{mooring}$ the vector pulling force in the three directions of mooring rope's mass point, while $T_{mooring}$ represents the scalar pulling force value generated by the stretching of adjacent mooring rope's mass point. $\mathbf{F}_{D-mooring}$ and $\mathbf{F}_{I-mooring}$ represent drag force and inertia force of mooring rope's mass point, respectively, $\mathbf{G}_{mooring}$ the gravity at the mass point of mooring rope, $\mathbf{F}_{float-mooring}$ the buoyancy of mooring rope's mass point; $L_0$ is the original length of the mooring rope, $l$ is the length after deformation. $C_1$ and $C_2$ are elastic coefficients of mooring rope's materials; $d$ is the diameter of the mooring rope.

The simulation method, which includes the tie point between the mooring rope and floating frame or between mooring rope and another mooring rope, needs special treatment. Considering that any tie point of mooring rope must be at the mass points of the head and tail of mooring rope, the physical quantity at the mass points of the head and tail of mooring rope is still stored in the storage array when the model is established. However, the mass points at the head and tail of mooring rope are only calculated once at a certain time. After all other mass points and units are calculated, find the corresponding tie points instead of saving the physical quantities of the head and tail mass points of mooring rope. In this paper, two mooring methods of floating frame are involved. Take Figure 3 as an example; one is to connect the four corners of the floating frame directly to the seafloor, the other is to tie the four corners of the floating frame with a small diameter auxiliary mooring rope, and then connect a large diameter mooring rope to the seafloor. In the first mooring method, the head and tail points of each mooring rope are in the same form, and the tail points are fixed on the sea bottom. The physical quantity of motion of the first point and the four corner's endpoints of floating frame are repeated, so they are not involved in the calculation in the model. When all the physical quantities of floating reef are calculated at a certain time, the physical quantities of the four corners of floating frame are directly assigned to the first point of the mooring rope. The second mooring method adopts two kinds of direct mooring ropes, large and small, and the tail points of four auxiliary mooring ropes are also the first points of main mooring rope. Therefore, in the model, the head and tail points of the four auxiliary mooring ropes and the tail points of the main mooring rope are not involved in the calculation, and the main points of the main mooring rope are selected separately for calculation, as shown in Figure 3b. After all the physical quantities of floating reef's movement are calculated at a certain time, the physical quantities of the four corners of floating frame are directly assigned to the first points of four auxiliary mooring ropes, and the physical quantities of the first point of main mooring rope are assigned to the first points of auxiliary mooring rope. The benefits of this treatment are mainly reflected in the same description of the constituent points of floating reef of each type, so it is easy to realize the model's horizontal topology and post-processing.

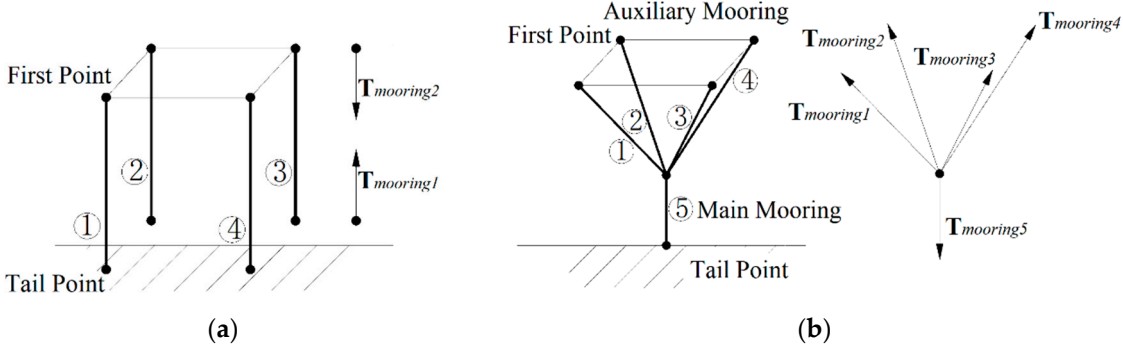

**Figure 3.** Calculation method of the two patterns of mooring. (**a**) Mooring mode and tension of four mooring rope. (**b**) Mooring mode and tension of four mooring rope of the auxiliary mooring rope matched with a main mooring rope.

### 2.4. Method for Judging the Floating Reef out of Water Surface

When the movement position of the artificial floating reef under the action of waves is at the trough, some calculated mass point or unit of floating reef may be above the water surface. In this condition, the mass points or units of the floating reefs will be free from buoyancy and wave forces. Therefore, the establishment of the numerical model requires the judgment of the floating reef out of water surface. The judgment method is the relationship between the $z$ coordinate of each calculated mass point or unit of the floating reef and the wave surface, as shown in Equation (13): when $z$ is greater than $\eta$, the velocity and acceleration of the water quality point are both to 0, and the buoyancy force of the calculated mass point or unit is 0 accordingly. When $z$ is less than or equal to $\eta$, the relative

water depth correction of wave surface change is considered. As shown in Equation (14), $z$ is modified to $zd$, and the calculation is carried out according to the linear wave theory [17]. The modified wave water mass point parameters at the $zd$ position are more consistent with the actual situation under the water outlet condition [18].

$$\eta = a\cos(ky - \omega t) \tag{13}$$

$$zd = d_{wave} \cdot \frac{z + d_{wave}}{\eta + d_{wave}} - d_{wave} \tag{14}$$

In the formula, the coordinate system is built on the static water surface, and the vertical upward axis is taken as $z$-axis. It is stipulated that waves propagate along the positive direction $y$. $\eta$ is the water surface variation, $a$ is the wave amplitude, $k$ is the wave number, $\omega$ is the circular frequency; $d_{wave}$ is the water depth, and $zd$ is the corrected position coordinate.

In particular, it is pointed out that, although the velocity and acceleration of water quality point at the position of each calculated mass point or unit are 0 under the condition of out of water surface, the relative velocity of mass point or unit is adopted in Morrison formula to calculate the dragging force of moving mass point or unit. However, the mass point or unit velocity at this moment is generally not 0, which requires further judgment and is forced to be 0.

## 3. Validation of Numerical Model

The main ways to obtain verification data are field data collection and physical model test. Since artificial floating reef designed in this paper has not been put into the sea field, the physical model test method is adopted to verify the correctness of numerical model verification. The motion state of floating reef is the result of the joint action of various forces under the action of waves, in order to facilitate and simplify the test, the mooring mode of four identical mooring ropes are selected and the floating reef does not out of water surface. The correctness of the numerical model is verified only by the motion change of a point on floating reef.

### 3.1. Physical Model Making

The physical model of floating frame needs to meet the geometric similarity, dynamic similarity and motion similarity. The stiffness similarity of the circular tube is ignored, and the circular tube is considered as a rigid structure in the prototype. In order to ensure the similarity of gravity and inertia forces between the floating frame model and the prototype, Froude number Fr should be equal [19,20]. Since the force and movement of floating frame under wave action have periodic changes, periodicity similarity should also be guaranteed. The Strouhal number is equal, as shown in Equation (15). In order to ensure the similar hydrodynamics of the mesh, the fishing net with larger mesh should be selected in the physical model making to avoid the problem of dissimilarity of water flow caused by too small mesh. The mooring rope mainly meets the elastic similarity, and rubber bands can be artificially added to increase the elasticity of the mooring rope model.

$$\begin{cases} \dfrac{V_m}{\sqrt{gL_m}} = \dfrac{V_p}{\sqrt{gL_p}} \\ \dfrac{V_m T_m}{L_m} = \dfrac{V_p T_p}{L_p} \end{cases} \tag{15}$$

where $V^*$, $L^*$, and $T^*$ are the characteristic velocity, characteristic line scale and period of the floating body structure, respectively. $m$ and $p$ represent models and prototypes, respectively. According to similarity criterion, the proportional relationship between the physical quantities of prototype and model can be deduced, as shown in Table 1. The floating frame is comprised of the PVC tubes, the joints, and the PVC glue. Except for the top and bottom sides of floating frame, fishing net is wound around the other sides, and mooring ropes are respectively tied to the endpoints of floating frame's bottom. The length,

width, and height of the wave tank are 32 m, 0.8 m, and 1 m, respectively. Comprehensive consideration of the size of the wave flume and the actual ocean area, $\lambda = 20$ was selected to establish physical model. The specific structural parameters and physical drawings are shown in Table 2 and Figure 4, respectively. In Table 2, $C_D$, $C_I$, $C_1$, $C_2$, etc. are the main parameters for calculating wave forces of floating frame, fish netting and mooring rope, and specific values refer to the research of Guo et al. [17], Wan et al. [18].

**Table 1.** Proportional relationship of parameters between prototype and model.

| Parameters | *p/m* | Parameters | *p/m* | Parameters | *p/m* |
|---|---|---|---|---|---|
| Length | $\lambda$ | Area | $\lambda^2$ | Volume | $\lambda^3$ |
| Period | $\sqrt{\lambda}$ | Frequency | $1/\lambda^2$ | Gravity | $\lambda^3$ |
| Angle | 1 | Linear velocity | $\sqrt{\lambda}$ | Inertia moment | $\lambda^5$ |
| Time | $\sqrt{\lambda}$ | Rotational inertia | $\lambda^5$ | Resistance | $\lambda^3$ |

**Table 2.** Physical model setting parameters.

| Parameters | Materials | Size (m) | Diameter (m) | Mesh Bar (m) | Density (kg/m³) | Coefficient of Drag Force $C_D$ | Coefficient of Inertia Force $C_I$ | Coefficient of Elasticity $C_1$ | Coefficient of Elasticity $C_2$ |
|---|---|---|---|---|---|---|---|---|---|
| Floating frame | PVC | $0.16 \times 0.16 \times 0.24$ | 0.01 | / | 646.57 | 0.8 | 1.2 | / | / |
| Fishing net | PE | $0.64 \times 0.24$ | 0.0015 | 0.048 | 953 | 0.6 | 1.2 | $345.3 \times 10^6$ | 1.0121 |
| mooring rope | PE | 0.25 | 0.001 | / | 953 | 0.6 | 1.2 | $345.3 \times 10^6$ | 1.0121 |

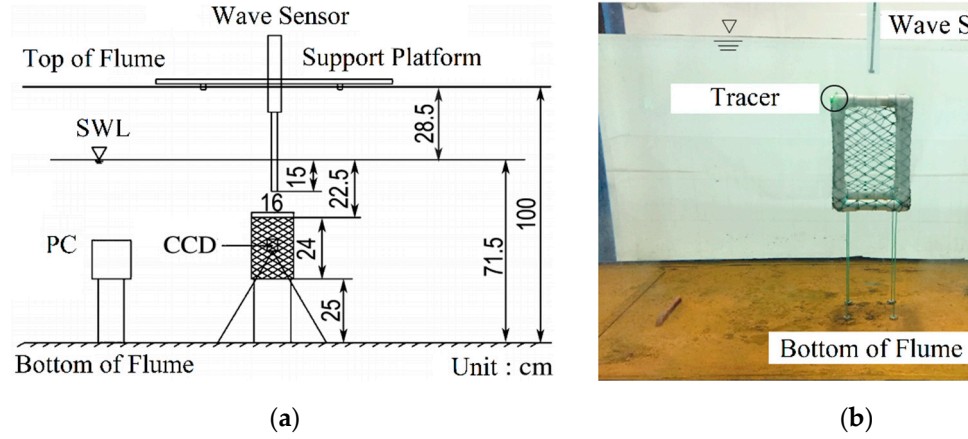

**Figure 4.** Experiment layout of a floating fish reef in a flume. (**a**) Arrangement of experiment instruments. (**b**) Physical model of floating reef.

### 3.2. Physical Modeling Test

The physical modeling test was carried out in a wave tank at the School of Marine Engineering Equipment, Zhejiang Ocean University, China. The length, width, and height of the wave tank are 32 m, 0.8 m, and 1 m, respectively. The end of tank adopts a pore structure to eliminate waves. The water depth is 715 mm, the wave period is set as 1.1 s, 1.4 s, 2.1 s and 4.0 s, respectively, and the corresponding wave height is 4.58 cm, 166.3 mm, 112.3 mm, and 183.6 mm. There are four groups of wave properties, as shown in Table 3. The wave properties were statistically measured by the wave sensor above floating reef model in Figure 4. CCD camera was used to collect the continuous images of the trace points in Figure 4, and a program was written to analyze the images and obtain the motion changes of the model. The frame rate of image acquisition will affect the accuracy of data analysis in the later period. The more frames, the more accurate the trace track will be, and the higher the accuracy of the test results. The requirement can be met if the frame

number is set at 10 per second. Therefore, the test recording time was 120 s and the number of frames was 10 per second. Each wave state was repeated three times in parallel.

**Table 3.** Parameters of wave cases and experiment results analysis.

| Wave Case | Period/s | Wave Height/mm | Physical Modeling Experiment | | | Numerical Simulation Calculation | | | Error [1] | | |
|---|---|---|---|---|---|---|---|---|---|---|---|
| | | | Maximum Negative Surging/mm | Maximum Positive Surging/mm | Maximum Heaving/mm | Maximum Negative Surging/mm | Maximum Positive Surging/mm | Maximum Heaving/mm | Maximum Negative Surging/mm | Maximum Positive Surging/mm | Maximum Heaving/mm |
| (1) | 4.0 | 45.8 | −28.2 | 21.1 | −3.6 | −19.9 | 20.1 | −1.0 | −8.3 | 1.0 | −2.6 |
| (2) | 2.1 | 166.3 | −71.2 | 75.3 | −14.8 | −59.0 | 70.3 | −9.9 | −12.2 | 5.0 | −4.9 |
| (3) | 1.4 | 112.3 | −63.2 | 65.0 | −10.8 | −52.6 | 58.9 | −6.9 | −10.6 | 6.1 | −3.9 |
| (4) | 1.1 | 183.6 | −43.9 | 45.7 | −6.7 | −33.6 | 39.5 | −3.1 | −10.3 | 6.2 | −3.6 |

[1] The error is the experiment value minus the numerical simulation value.

Before wave building, the floating reef was in a static state, and the position of floating reef model at this time was selected as the reference standard. MATLAB is used to mark the positions of the tracer points at different times on the same picture to get the track of the tracer points. Use a vernier caliper to measure the length of a circular tube of the floating frame, and calculate the number of image pixels of the circular tube at motionless condition in the water, and then obtain the actual length of a pixel space. Choose different circular tubes of the floating frame and repeat the calculation several times. After taking the average, the length of one pixel in the image is 0.448 mm. First, through processing the CCD black and white image gray scale, the gray scale images of gray scale value of each pixel were obtained. Gray scale value range from 0 to 255. Black is 0 and white is 255. There are pixels with 254 or 253 around the white point. This article only calculates the pixels with a pixel gray value of 255 [21].

*3.3. Comparison and Verification of Results*

The numerical model of the floating reef is established according to the proportion relationship among parameters in Table 1. The trajectories of the tracers are calculated and then reduced to the scale of the physical model with $\lambda = 20$. Figure 5 shows the change of wave surface, which is acquired synchronously by the wave sensor and the CCD acquisition system. The still water surface is the base plane of the Y-axis. It can be seen that the wave case (1) in Figure 5 is a linear wave. As the wave height and frequency are increased under the condition of constant water depth, the wave trough will appear attenuation, so the wave case (2), wave case (3), and wave case (4) are nonlinear waves. The maximum wave height appears when wave case (4) occurs. This is because the end of the tank cannot completely eliminate the reflected wave that slightly affects the periodicity of the wave case (4) surface. Figure 6 shows the location and trajectory comparison of the tracer points in the physical model test and numerical simulation, and Table 3 shows the comparison of the maximum positive surging, negative surging and the maximum heaving. In particular, the origin of the coordinates in Figure 6 is the position of the tracer when the floating reef is stationary. Define the right side of the tracer point as the positive direction of the X axis, and the upper side as the positive direction of the Y axis. The origin is 225 mm below the still water surface.

According to Table 3 and Figure 6, by comparing the results of physical model test and numerical simulation, it is found that the movement result of physical model test is larger than that of numerical simulation in horizontal direction. In the vertical direction, the verification of the left part is better, and the numerical simulation results of the right part are greater than the physical model test. With the depth of the wave as the characteristic length, the maximum deviation error is between −12.2 mm and 6.2 mm in surging and heaving. The main reasons that cause the error rate can be summarized as the following: Firstly, the limited water depth of the wave tank. Secondly, the measured wave in the condition of large wave height is a nonlinear wave, but the linear wave theory is used in the numerical model, which needs further refinement and improvement. Certainly, there are other reasons not considered in the numerical model, such as the reflection phenomenon

of wave making in the water tank for a long time, and the use of thin iron wire to tie the fishing net in the physical model. Considering comprehensively, the established numerical model can describe the movement of the floating reef under the action of waves, and also indirectly verify the force of fishing net and mooring rope.

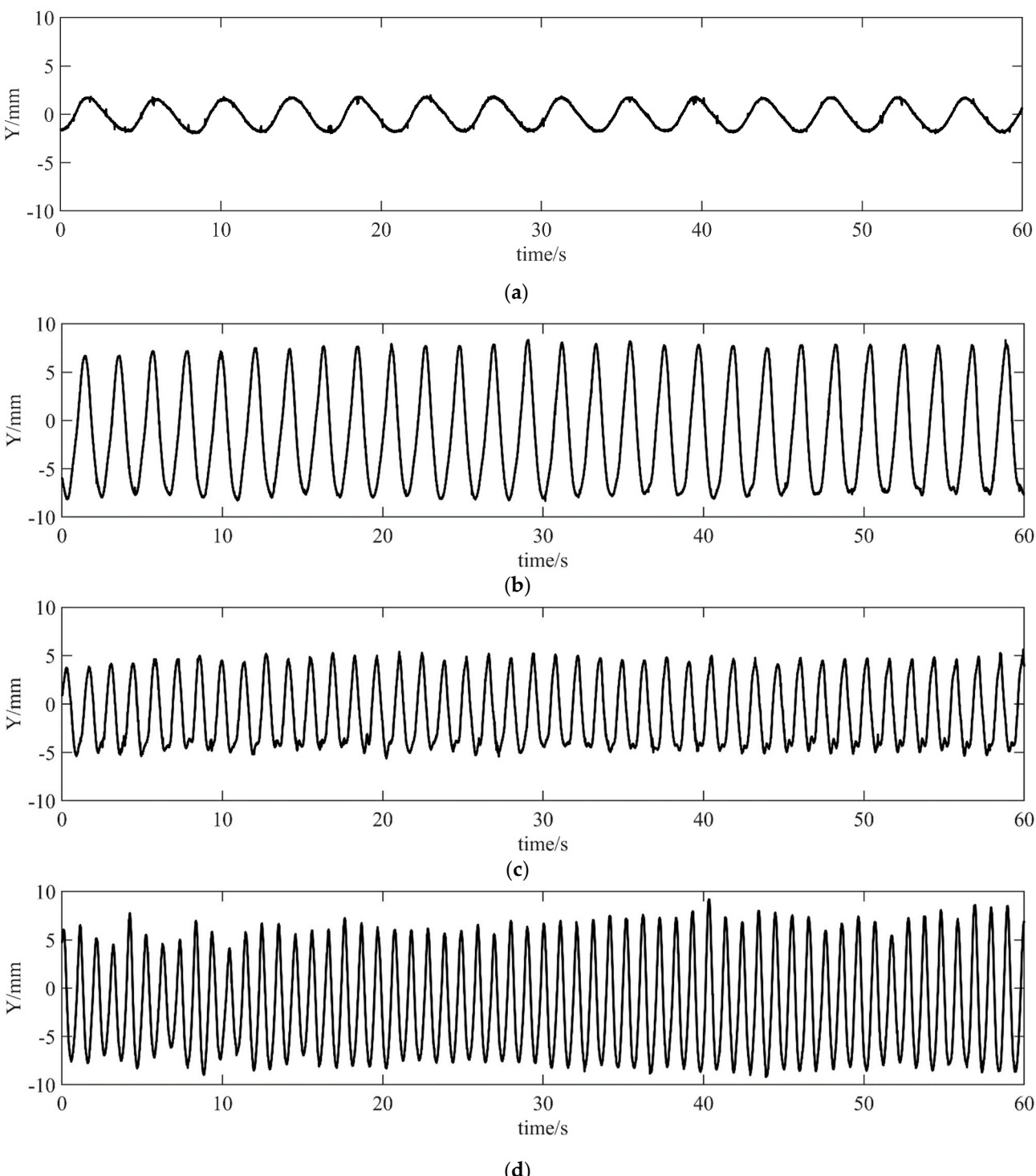

**Figure 5.** Four group wave surface changes within 60 s collected by Wave sensor. (**a**) Wave case (1); (**b**) Wave case (2); (**c**) Wave case (3); (**d**) Wave case (4).

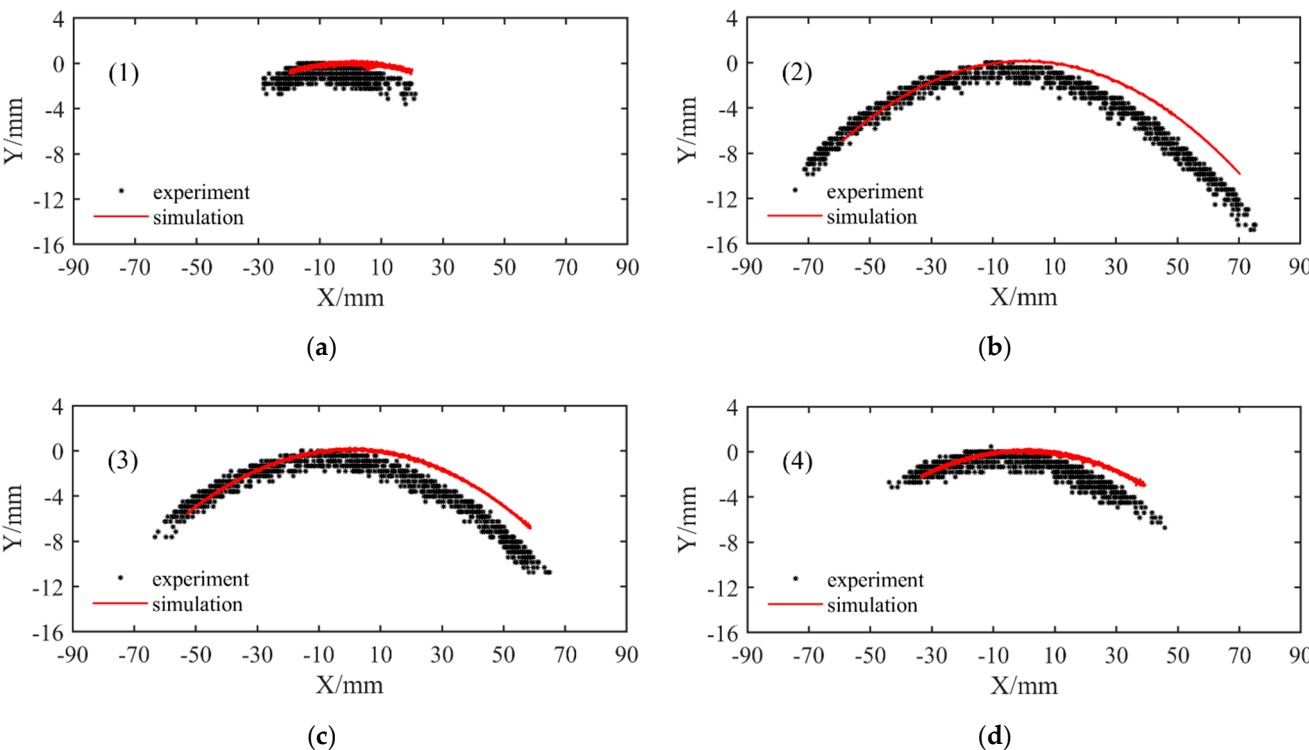

**Figure 6.** Movement track of the trace point of physical experiment and numerical mode under four group waves. (**a**) Wave case (1); (**b**) Wave case (2); (**c**) Wave case (3); (**d**) Wave case (4). The origin of the coordinates is the position of the tracer when the floating reef is stationary.

## 4. Results and Discussion

The artificial floating reef may be out of the water surface under the action of waves, so the movement and force change of the actual size of floating reef are calculated according to the numerical method described in Section 1, where the mooring is the combination of the auxiliary mooring and the main mooring as shown in Figure 3. According to the parameters in Table 2, the numerical model size of the floating reef is enlarged to its actual size by $\lambda = 20$. The size of the float frame is 1.69 m × 1.69 m × 4.8 m, and the diameter of the circular tube is 0.2 m. Otherwise, the density is the same as described above. The diameter of fishing net is 0.003 m, and mesh bar is 0.32 m. The diameter of the main mooring rope is 0.02 m, and the diameter of the auxiliary mooring is 0.01 m. In Figure 3, the distance between the middle node of mooring rope and the bottom center of floating frame is 1 m. Other physical parameters and calculated parameters are the same as those in Table 2. According to the statistical wave properties of 5% cumulative frequency in Zhoushan sea area in China [22], the wave height is 3 m and the water depth is 12 m. The wave steepness is 1/20, and the wave period calculated according to the dispersion equation is 6.7 s, which is close to the wave period 6 s of 5% cumulative frequency as calculated in literature [22].

The duration curve of three factors which include the pitching angle of floating frame, the tension of the mooring rope, and the total tension of the tying point of fishing net are shown in Figure 7. In particular, this paper makes the following explanation: Firstly, the wave direction is positive and the opposite direction is negative. Secondly, the tension of the mooring rope and the total tension of tie points of fishing net within 20 s after 50 s calculation show that the movement of the floating reef has stabilized in the numerical model. Last but not least, it can be concluded that the three factors have obvious periodic changes, and the maximum rope tension and the maximum total tension at the tie point all appear at the position where the pitching angle suddenly changes. At the position marked by the circle in Figure 7, the negative angle of the pitching angle increases and suddenly stops, which cause the tension of the mooring rope and the total tension of the tie

point to reach the maximum and change disorderly. In Figure 7, the rectangular moment represents the maximum pitching angle of floating reef, and the tension of mooring rope is the smallest at this moment, which is opposite to the variation trend of the total tension of tie points. Figure 8 shows the motion state of the floating reef and the force distribution of the fishing net at the moment of maximum pitching angle. The force of the fishing net is only on the wave's front and back, and the tension value is far less than that of mooring rope. Therefore, the smoothness of the duration curve of the pitching angle of floating reef can be used as the basis for optimizing the structure shape of floating reef and matching floating body and weight.

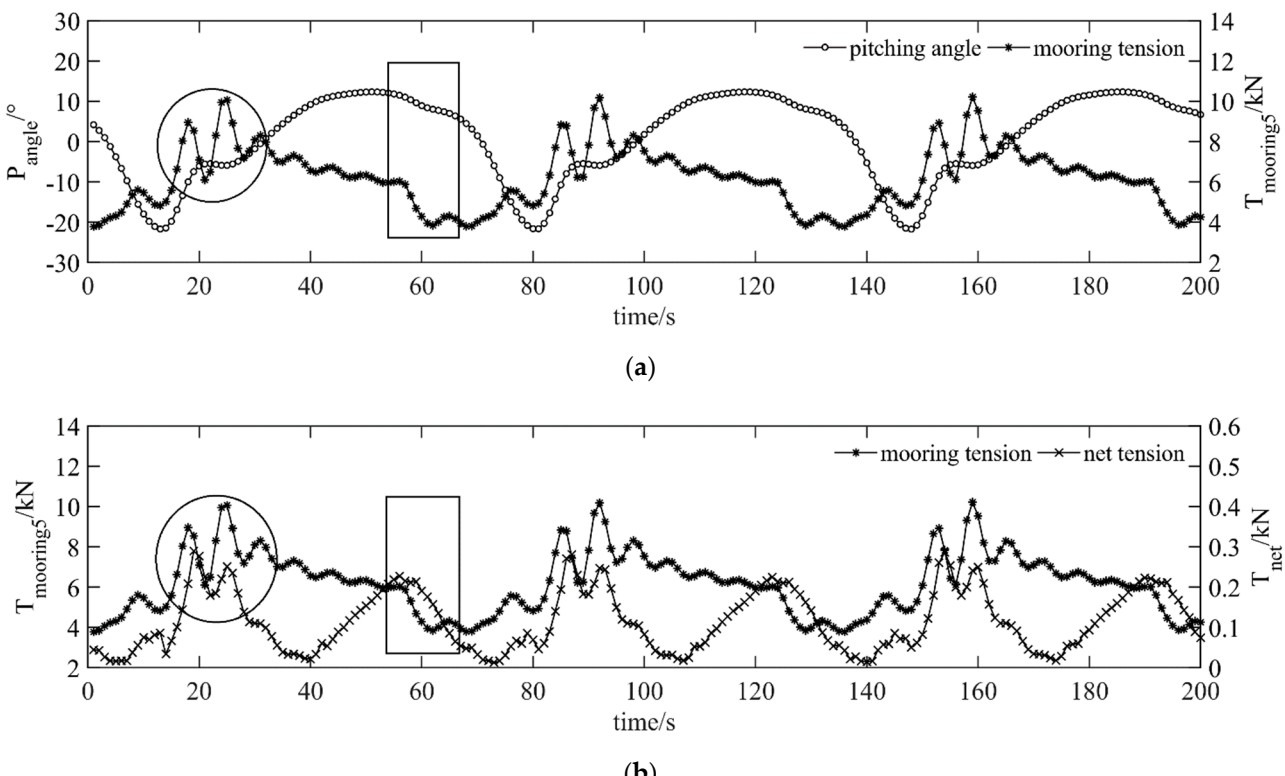

**Figure 7.** During curve of the pitching angle of floating reef, the tension of mooring rope, and total tension of fishing net's tie points. (**a**) The relationship between the pitching angle of floating reef and the tension of mooring rope. (**b**) The relationship between the tension of mooring rope and total tension of fishing net's tie points.

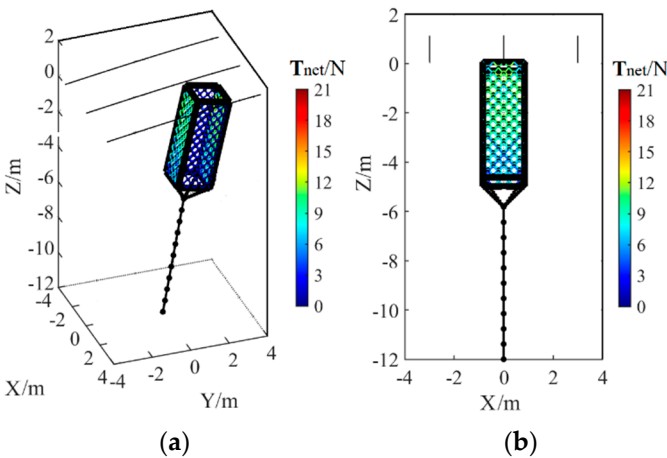

**Figure 8.** The motion state of the floating reef and the force distribution of fishing net at the maximum pitching angle. (**a**) 3D view. (**b**) Front view.

## 5. Conclusions

Based on the finite element method and the lumped mass point method, the simulation of the forces and motions of floating reef were calculated. For detail, the method for processing coupling forces and motion, the method for judging the floating reef out of water surface, and the method for correcting velocity and acceleration of water quality points are elaborated in detail by using finite element method and lumped-mass mooring model. Finally, a numerical model of the dynamic response of floating reef with the combination of the auxiliary mooring and the main mooring is established. By analyzing the duration curves which include pitching angle of floating reef, the tension of mooring rope and the total tension of tie points of fishing net, the main conclusions are summarized as follows:

1.  The key to the establishment of the hydrodynamic numerical model of floating reef is the unit division and spatial topology of its components. In this paper, a topological approach integrating the floating frame, the fishing net, and the mooring rope was given in a top-down order. In addition, the calculation method of floating reef's motion and force was also given in this paper. The modeling method is applicable to the analysis of hydrodynamic characteristics of rigid and flexible structures with small size or complex structures composed of rigid and flexible structures.
2.  The tension of mooring rope of floating reef under single mooring conditions is related to the movement of artificial floating reef. The sudden change of the pitching angle will cause the tension of mooring rope and the total tension of tie points to reach the maximum value and change disorderly. Therefore, the smoothness of the duration curve of the pitching angle of floating reef can be used as a basis for floating reef to optimize the structure and to match structural shape and counterbalance.
3.  Under the condition of single mooring rope, the surfaces of force of the fishing net of floating reef are mainly seen on the wave's front and back, and the total tension of the tie points is much smaller than the tension of mooring rope. Therefore, the floating reef winding fishing net only adds a small amount of load while forming a closed hollow cube, which is the preferred form of the design of floating reef.

**Author Contributions:** Conceptualization, methodology and writing—original draft preparation, Y.P. and Y.Z.; writing—review and check, H.T.; project administration and funding acquisition, C.L.; visualization, D.X. All authors have read and agreed to the published version of the manuscript.

**Funding:** This research was funded by the National Natural Science Foundation of China (No. 42006175), the Zhejiang Provincial Natural Science Foundation of China (No. LQ19E090007, No. LQ20E090004, No. LQ18E090007).

**Institutional Review Board Statement:** Not applicable.

**Informed Consent Statement:** Not applicable.

**Data Availability Statement:** Not applicable.

**Acknowledgments:** We would like to thank the financial support of the National Natural Science Foundation of China(No. 42006175). We also acknowledge that this study was partially funded by the Zhejiang Provincial Natural Science Foundation of China (No. LQ19E090007, No. LQ20E090004, No. LQ18E090007). The authors would like to express profound thanks to them.

**Conflicts of Interest:** The authors declare no conflict of interest.

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
