# Peer review of "Numerical Simulation Study on Environment-Friendly Floating Reef in Offshore Ecological Belt under Wave Action"

_water, doi:10.3390/w13162257_

Round 1
Reviewer 1 Report
The paper concerns with a numerical study for the evaluation of the forces acting on an artificial floating reef subjected to wave action.
Comments
From a general point of view, the paper is well organized and the results all well described. Nevertheless, since the results are obtained by a numerical approach, the authors should better describe some numerical aspect of the model. In particular, the authors should explain in detail how the wave action exerted on the artificial reef has been evaluated and numerically simulated.
Reviewer 2 Report
This paper is dealing with a numerical simulation of force and motion of floating reef. The paper is showing interesting results for practical applications under wave motion. However, in order to be published in an international journal, the paper needs to be revised much more considering the following items.
- Figure 5 (movement track) : The definition of the variable (X,Y) must be clearly defined, by showing its definition in Figure 4.
- As compared with an explanation for the numerical model in “2. Establishment of numerical model,” highly limited description is given for the laboratory experiment. Further explanations are required for (i) the length of the wave flume, (ii) details of wave absorption method and reflection coefficient in the wave flume, as the authors mention that the discrepancy in Figure 5 is attributed to wave reflection, and (iii) wave form generated in the flume to show to what extent it is similar to or deviates from sinusoidal time-variation. As seen in Eq. (9), the waveform is simply assumed to be sinusoidal in the present analysis. However, if the waveform in the present experiment is much different from sinusoidal, it may cause the deviation from the simulation result as illustrated in Figure 5, although the authors simply stressed that the reason of deviation in Figure 5 is induced by wave reflection in the flume.
- Figure 5: Why does the large deviation appear only for the positive values? Meanwhile, the negative values show perfect agreement. The author expects the wave non-linearity might be related to this deviation characteristic (see the comment (2) above).
- Table 3 : Write the definition of the relative error clearly by mathematics. In the text, the authors wrote “choosing water depth to calculate the relative error.” However, it is a strange definition as an error. I strongly suggest you to use normal error, instead. If you don’t, please write more detailed reason why use it.
Round 2
Reviewer 2 Report
The paper has sufficiently been improved in response to Reviewer's comments.